# MyoChallenge 2023
# Towards Human-Level Dexterity and Agility

**Vittorio Caggiano**[1*]   **Guillaume Durandau**[2*]   **Cheryl Wang**[2*]   **Chun Kwang Tan**[3*]
**Pierre Schumacher**[4*]   **Huwawei Wang**[5]   **Alberto Chiappa**[6]   **Alessandro Marin Vargas**[6]
**Alexander Mathis**[6]   **Jungnam Park**[7]   **Jungdam Won**[7]   **Gunwoo Park**[8]   **Beomsoo Shin**[8]
**Minseung Kim**[8]   **Seungbum Koo**[8]   **Zhuo Yang**[9]   **Wei Dang**[9]   **Heng Cai**[9]   **Jianfei Song**[9]
**Seungmoon Song**[3]   **Massimo Sartori**[5]   **Vikash Kumar**[1,10]

[1]MyoLab    [2]McGill University and Jewish Rehabilitation Hospital
[3]Northeastern University    [4]Max Planck Institute for Intelligent Systems
[5]University of Twente    [6]Ecole Polytechnique Fédérale de Lausanne (EPFL)
[7]Seoul National University    [8]Korea Advanced Institute of Science and Technology (KAIST)
[9]Carbon Silicon AI    [10]Robotics Institute, CMU

## Abstract

Humans move nimbly and with ease, capable of effortlessly grasping items of many shapes and qualities. Over millions of years, the musculoskeletal structure, central and peripheral neural systems have evolved together to provide this capacity. Understanding the underlying mechanisms of this complex system helps translate benefits to other fields, from robot locomotion to rehabilitation. To illicit new insights into the generation of diverse movements and precise control as well as foster collaboration between the biomechanics and the ML community, the *MyoChallenge* at the NeurIPS 2023 Competition featured two tracks: Manipulation and Locomotion. Manipulation involved precisely manoeuvering an object of varying shape by controlling a 63-musculoskeletal arm model and generating stable grasps. Locomotion involved the combination of abstract reasoning and low-level control, as agents have to chase or evade from a moving object by controlling an 80-musculoskeletal model of human legs. These tasks best highlighted our overarching theme of dexterity and agility, requiring the generation of skilled and efficient movements with realistic human limbs. The Myosuite framework enabled the challenge through a realistic, contact-rich and computation-efficient virtual neuromusculoskeletal model of the human arm and legs. This was the second iteration of the MyoChallenge with 59 teams participating, and over 500 submissions. Each task involved two phases, increasing in difficulty over time. While many teams achieved high performance in phase 1 for the Manipulation track, locomotion showed variable performance across participants. In phase two, scores for all teams dropped significantly as the focus shifted towards generalization under uncertain conditions, highlighting the need for stronger generalization in agents In future challenges, we will continue to pursue the generalizability in dexterous manipulation and agile locomotion, which is crucial for understanding motor constructs in humans.

**Challenge Webpage**: https://sites.google.com/view/myochallenge

---

[*]co-first

Submitted to the 38th Conference on Neural Information Processing Systems (NeurIPS 2024) Track on Datasets and Benchmarks. Do not distribute.

# 1  Introduction

The excellence of humans in performing complex and highly agile movements is fundamentally linked to the nuanced and simultaneous control of various muscle groups. Our musculoskeletal system, composed of bones of differing lengths connected by an array of skeletal muscles, tendons and other types of connective tissue, is an extremely complex biological system, resulting from millions of years of evolution. The neuromuscular structure that governs this system operates within a high-dimensional space, involving approximately 600 muscles coordinating around 300 joints [1]. This system's redundancy, where multiple muscles can act on a single joint, and its multi-articular nature, where a single muscle may influence multiple joints, are critical for the versatility and efficiency of our movements. However, this complexity comes at a cost: it is still not understood how the brain controls all aspects of the neuro-musculoskeletal system

Modeling human motor control poses a significant scientific challenge with wide-reaching implications across numerous fields, including neuroscience, biomechanics, ergonomics, assistive robotics, and rehabilitation medicine. The development of various models has been instrumental in understanding motion control, yet many remain abstract and do not fully capture the complexities of how movements are generated [2, 3, 4]. Moreover, musculoskeletal models are typically designed for specific tasks, which restricts their applicability and scalability to more complex or diverse actions. Furthermore, while neuromechanics models and simulations serve as vital platforms for testing control theories and illustrating motion production through physiologically plausible musculoskeletal dynamics, there remains a significant gap in creating models that are versatile, adaptable, and generalizable for both manipulation and locomotion domains. Bridging this gap is crucial for advancing our understanding and enhancing the practical applications of human motor intelligence, aiming to develop models that accurately reflect the sophisticated nature of human movement.

In recent years, significant advancements in the fields of biomechanics, machine learning [5, 6, 7], neuroscience, and physics simulators [8, 9, 10, 11] have been observed. However, these disciplines have largely evolved independently. In order to leverage new developments in algorithmic control and complex learning architectures to further our understanding of human motor control, MyoChallenge was launched while seeing an opportunity to bring together experts from these varied fields to enhance understanding of human motor control. This renewed approach was motivated by the desire to leverage state-of-the-art simulators and machine learning techniques. The aim is to address the existing gap by creating models that are not only versatile and adaptable but also generalizable across both manipulation and locomotion domains, thus pushing the boundaries of what is currently possible in modeling human movement. Specifically, the question that we want to address with this challenge is: *Can we match human level dexterity and agility with physiological digital twins?*

Building on the NeurIPS 2022: MyoChallenge's success [12], *MyoChallenge* 2023 proposes two unique challenges, one: to control a realistic musculoskeletal arm model for a more complex manipulation task, and two: to control a musculoskeletal leg model in a chase/evade task, inspired by the ChaseTag game [13].

To handle the above complexity, *MyoChallenge* leverages MyoSuite[2] - an open-source framework that implements highly efficient computational biomechanical models and allows muscle-driven simulations of these models to solve skilled tasks [14]. MyoSuite offers physiologically accurate musculoskeletal full hand models [15] in a framework that is several orders of magnitude (up to 4000x) (see Figure 7 in [11]) faster than the state of art musculoskeletal simulators [16, 17] used in previous challenges. MyoSuite also support full contact dynamics, which most competing alternatives lack, to enable contact rich manipulation behaviors.

---

[2]https://sites.google.com/view/myosuite

## 2 MyoChallenge 23': Task and evaluations

Modeling human motor control to produce human-like, versatile, adaptable, and generalizable manipulation and locomotion has far-reaching implications in neuroscience, biomechanics, assistive robotics, and rehabilitation medicine. However, a significant gap still exists between current neuromechanical simulations and biomimetic behaviors. Extending from MyoChallenge 22', we present a competition track in *MyoChallenge 23'* that requires control of full arm movement with multiple-object manipulations and lower-limb locomotion tasks. Here, we present the rationale behind the tasks (Sec. 2.1), the *MyoArm* and *MyoLeg* model (Sec. 2.2), and finally the tasks proposed (Sec. 2.3).

### 2.1 Design philosophy

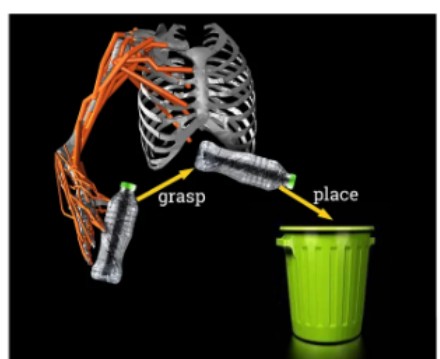
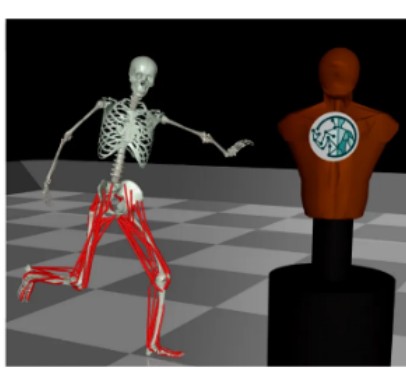

Figure 1: Two tracks of MyoChallenge 2023: **A.** the manipulation track where a full musculoskeletal arm model will be reaching, grasping, controlling, and moving a real object to achieve a goal and **B.** the locomotion track, where a bilateral musculoskeletal leg model will be controlling a human body to chase or evade a moving target.

This year's MyoChallenge consists of two distinct tracks focusing on manipulation and locomotion.

**1. Manipulation Track** (Fig.1-A), presents a task of reaching to grasp and properly maneuver an object to move it to a target location of the workspace. This task entails the high complexity of single arm-hand dexterity to manipulate the surrounding objects to achieve the goals of moving and placing.

**2. Locomotion track** (Fig.1-B), represents the locomotion task to chase or evade a moving goal with a musculoskeletal model of the lower limbs. The complexity of this task is within the nimble and agile dynamic control and decision-making in the lower body.

### 2.2 Musculoskeletal Arm and Leg Models

The manipulation track uses the *MyoArm*, a neuromusculoskeletal model representing the torso and the right arm consisting of 63 muscles and 27 internal DOFs. The locomotion track uses *MyoLeg* to present the whole body with articulated legs, consisting of 80 muscles and 16 internal DOFs. This model is based on [18] and follows its definitions and conventions. Both models feature a skin layer that enables full contact with the environment.

### 2.3 Tasks

The participants could participate in either track, consisting each of two phases with increasing difficulties and randomization.

#### 2.3.1 Manipulation Tasks

In this track, the participants were asked to develop a general manipulation policy capable of interacting with common household objects, such as children's toys. The action space consists of a

63-dimensional vector representing the muscle stimulation signals of the MyoArm. The observation
state space is a vector containing the kinematic and muscle states of MyoArm and the object state.

In Phase 1, the task focuses on training a policy capable of picking up a specific object and manipulating it toward a receptacle bin with randomized orientation and position. Goals were randomly sampled to assess the generalization capabilities of the acquired behaviors. The second phase involved applying the policy to objects with new geometries and physical properties (e.g., mass and friction). Additionally, the object and MyoArm's initial configuration were randomized.

| Task - Phase | Position [mm] | Orientation [rad] | Size (L,W,H) [m] | Mass [kg] | Friction Coefficient |
|---|---|---|---|---|---|
| Relocate - 1 | ± 10 | ± 1.57 | (0.0284, 0.0284, 0.0284) | 0.18 | (1.0, 0.005, 0.0001) |
| Relocate - 2 | ± 20 | ± 3.14 | (0.02, 0.02, 0.02) ± 0.005 | 0.175 ± 0.125 | ± (0.2, 0.001, 0.00002) |

Table 1: Summary of task variations for Manipulation track

### 2.3.2   Locomotion Tasks

The task for locomotion resembled the World Chase Tag competition, the MyoLeg musculoskeletal model is required to chase or evade an opponent in a 12 x 12-meter arena, known as the Quad. The participants were asked to develop policies that control the MyoLeg to efficiently navigate the environment to avoid or pursue an opponent during each 20-second round. The action space is an 80-dimensional vector representing the muscle control signals of the MyoLeg and the observation state consists of information on kinematic, ground reaction force, and muscle states of the MyoLeg, the opponent's location information, and the Quad map.

In Phase 1, the task focuses on training the agent to pursue an opponent within a 20-second timeframe on a plain Quad. The opponent's behavior varied from remaining stationary to actively running away from the agent across different rounds. During the second phase, the agent alternated between chasing and evading the opponent and the terrain of the arena changed randomly into uneven grounds. In the evading task, the agent had to avoid the opponent as long as possible without leaving the arena.

| Task - Phase | Task [Prob] | Terrain Height [m] | Opponent behavior [Prob] | Opponent velocity range [m/s] |
|---|---|---|---|---|
| Chasetag - 1 | Chase [1] | Flat [0] | Stationary [0.55] Random [0.45] | Stationary [0] Random [0 ± 2] |
| Chasetag - 2 | Chase [0.5] Evade [0.5] | Flat [0] Hills [0.13 ± 0.1] Steps [0.2 ± 0.1] Rough [0.075 ± 0.025] | Stationary [0.45] Random [0.35] Repeller [0.2] | Stationary [0] Random [0 ± 2] Repeller [0.65 ± 0.35] |

Table 2: Summary of task variations for Locomotion track

### 2.4   Submissions and Evaluation

In order to succeed, participants needed to obtain the highest success (in terms of goal achievement) with the minimum effort (in terms of lowest overall muscle activation) for manipulation. In locomotion, the participants are ranked based on both chase duration (in seconds) and highest success. The EvalAI platform (https://eval.ai) was used for hosting the challenge and to run the evaluation.

**Evaluation Metrics**. The manipulation task used a negative distance error $D_{t=H} = -|X_t - X_{goal}|$ at the end of the task horizon as a performance metric. Additionally, a physiological metric calculated from the total muscle activation was used to estimate metabolic power. The teams scoring above 90% [3] were instead ranked based on the physiological effort to encourage less muscle activations. The locomotion tasks used a score based on chase/evade duration (in seconds): $s = 1 - \frac{t}{T}/s = \frac{t}{T}$.

---

[3]Note that this threshold changes to 30% during the second phase

Additionally, another performance metric *Points* was used based on the number of successful tags over 100 evaluation episodes.

For quantitative evaluations of the submissions, participants were asked to upload their behavior policies to Eval AI which automatically evaluated them and updated results on a score-board. Final scores were averaged over multiple seeds and task variations.

# 3 Solution strategies

In this section, we describe the methods employed by the top three participating teams in each track. Imitating movements from a dataset is one of the common training paradigm for the Locomotion track, where all winning teams trained their policies using datasets of human-like movement to produce gait. Curriculum learning is also commonly observed in both tracks, as teams used this method to shape the way their policies learn. In this Challenge, we noticed a novel method to constrain policy exploration, which allowed one team to clinch the top place in the Manipulation track.

## 3.1 Manipulation Track Approach

### 3.1.1 Team Lattice (FIRST)

Team Lattice comprising Alberto Chiappa, Alessandro Marin Vargas, and Alexander Mathis from EPFL, emerged as the first-place winners. Their solution was the result of four key ingredients: on-policy reinforcement learning with Recurrent PPO ([19, 20]), latent exploration via Lattice ([21]), curriculum learning and domain randomization ([22]). Code is available at [4].

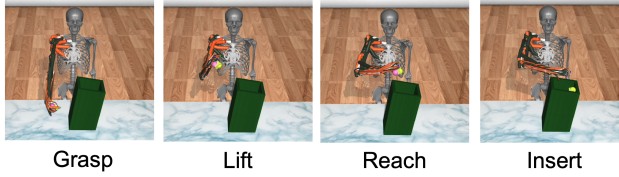

Figure 2: Curriculum steps (Lattice - Manipulation).

**Network architecture.** To address the partial observability of the environment, the team included an LSTM layer ([23]) before the two fully-connected layers of the policy network. In this way, the policy had the potential to keep in memory the inaccessible features of the environment, such as the shape of the object, inferring them from the transition dynamics.

**Lattice exploration.** To improve exploration, the stochastic policy followed a Lattice distribution, a multivariate Gaussian whose covariance depends on the learnt policy weights. In contrast to the original implementation of the exploration method ([21]), the exploration was modified for this challenge to sample actions in a state-independent manner, improving computational efficiency.

**Curriculum learning.** The agent was trained via a curriculum of task of increasing complexity (Figure 2). First, the agent learnt how to grasp the object with all the fingers. Second, the agent learnt to lift the object after grasping it. Third, the target was positioned above the receptacle. Fourth, the target was positioned inside the receptacle.

**Domain randomization.** To improve the robustness of the policy to unknown object shapes and environment conditions, the team widened the range of values from which the environment parameters (object size, mass and friction) could be sampled.

Finally, the team designed an early stopping criterion after which the agent would output minimum muscle activation thereby limiting the energy consumption. The early stopping criterion was designed to identify when the object has already reached the target location or when there is no hope to successfully place the object in the receptacle in time.

### 3.1.2 Team GaitNet (SECOND)

---

[4]`https://github.com/amathislab/myochallenge-lattice`

The second place in the Manipulation Track is Team GaitNet, with members consisting of Jungnam Park and Jungdam Won from Seoul National University. They used deep reinforcement learning (DRL) to train a controller with proximal policy optimization (PPO) [19] to move the MyoArm to desired locations. By looking at the object's initial position, goal position, and relative orientation, Team GaitNet proposed an **object trajectory generator.** By defining four initial key positions (blue circles in Figure 3), the generator produces the object's position $\hat{p}(t)$ as a function of time (red circles in Figure 3). The agent is then rewarded for correctly following the predefined target trajectory at each time step. Additionally, their reward function differentiates conditions between objects within and outside the

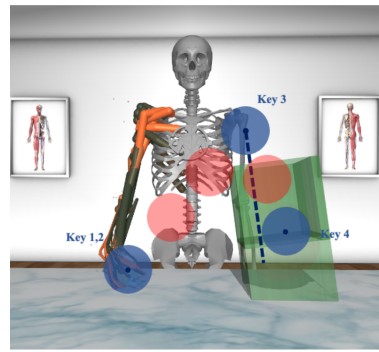

Figure 3: Training methods (Gait-Net - Manipulation).

box to encourage grasping and releasing the object at appropriate timesteps. The episode is truncated if the reward value doesn't meet a specific threshold value for learning efficiency, as proposed by [24].

### 3.1.3 Team CarbonSiliconAI (THIRD)

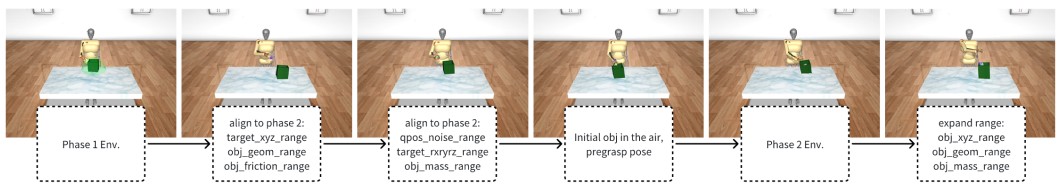

Figure 4: Curriculum steps (CarbonSiliconAI Manipulation)

The third place in the Manipulation track is Team CarbonSiliconAI, a team from CarbonSilicon AI Technology Co. Ltd. in Beijing, China. The team utilized Proximal Policy Optimization (PPO)[19] for curriculum learning, gradually increasing the difficulty of the task. As depicted in Figure 4, they aligned the model from the first phase environment to the second phase environment through multi-step curriculum learning, enabling a smoother transfer of prior experiences. It is worth noting that in the Phase 2 environment, objects are initialized in the air, which presents a more challenge for learning compared to alterations in shapes or physical parameters. To address this, they initialize the objects in the air and ensure that the palm is sufficiently close to the object before aligning to the Phase 2 environment, resulting in a more easily achievable pre-grasp posture. Additionally, they intensified the task difficulty based on the second phase environment by expanding the range of object properties (object location, size, and mass), resulting in improved performance of the model on edge cases.

## 3.2 Locomotion Track Approach

### 3.2.1 Team GaitNet (FIRST)

The winner of the Locomotion track is Team GaitNet, from Seoul National University, Korea, comprising of two members, Jungnam Park and Jungdam Won. The team employed a three-stage approach to train their policy. Proximal Policy Optimization (PPO) [19] was used to train their policies.

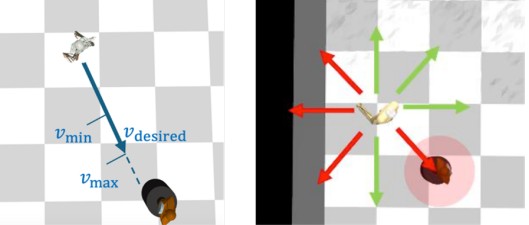

Figure 5: Desired velocity generation for the Chase task (Left) and the Evade task (Right).

**First stage.** The goal was to mimic walking mocap data in various directions. They selected eight motion clips from the Mixamo dataset [25], which include walking at 45-degree rotation intervals. Their controller was trained on three rewards,

given by $r = r_{\text{imit}} \times r_{\text{vel}} \times r_{\text{act}}$, where $r_{\text{imit}}$ denotes how well the policy matches the joint angles and relative feet positions from the center-of-mass, $r_{\text{vel}}$ denotes how well the policy matches a desired velocity, and $r_{\text{act}}$ rewards the policy for avoiding large changes in actions. The training was performed over all the terrain types provided by MyoChallenge.

**Second stage.** The goal is to move towards any given velocity. In this stage, they removed $r_{\text{imit}}$ from the reward function used in the first stage and added $r_{\text{face}}$ to align the agent's velocity and facing direction with the desired values. This allowed the controller to learn a variety of walking motions unrestricted by mocap data. They reported that the controller learned agile turns and other movements not presented in the mocap data during this stage.

**Third stage.** For the Chase task, the desired velocity was computed using the direction from the agent to the opponent as shown in Figure 5. For the Evade task, candidate velocities were first generated at 2-degree intervals from the agent, excluding velocities that move towards the boundaries or the opponent (see the red arrows in Figure 5). They then evaluated the value function (from the stage 2) for all the remaining velocities and selected the velocity with the highest value as the desired velocity.

### 3.2.2 Team MSKBioDyn (SECOND)

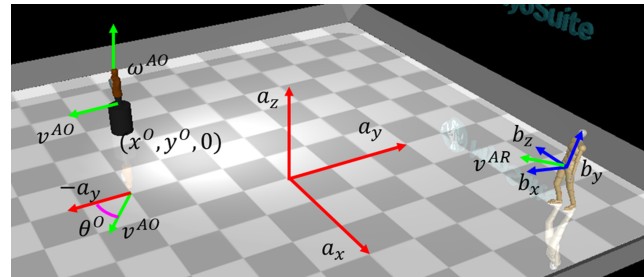

Figure 6: Model observations on the global and local reference frames. (MSKBioDyn)

The second place in the Locomotion track comes from Team MSKBioDyn, a team from KAIST, comprising Gunwoo Park, Beomsoo Shin, Minseung Kim, and Seungbum Koo. Their strategy involved training two multi-layer perceptron policy networks with PPO [19], one dedicated to chasing and the other to evading. Initially, only task rewards, calculated from the model's heading direction and velocity, were applied. However, the team noticed that the model could not walk robustly or realistically without prior knowledge of human motion. Accordingly, the style reward from adversarial motion priors [26] was applied to guide the agent in generating motion within the database. The comprehensive dataset required for the task comprised human motion clips of walking, running, and standing from the [27]. This data was converted into generalized coordinates for the MyoLeg model using inverse kinematics calculations in the OpenSim software [28]. The given observations for the global frame (red) were transformed into data for the local frame (blue) to reduce redundancy in learning (Figure 6). Since the team considered that kinematics and kinetics of the skeletal model would include information about muscle variables, muscle observations were not used for training. Although the agent could achieve some tasks without rewards based on muscle activation, the effect of activation minimization was not tested. Lastly, the agent was trained on tasks with increasing difficulties, from the level surface to full-scale terrain, via curriculum learning [29].

### 3.2.3 Team CarbonSiliconAI (THIRD)

The third place in the Locomotion track is Team CarbonSiliconAI, a team from CarbonSilicon AI Technology Co. Ltd. in Beijing, China. The team applied the two-stage framework (pre-training and task training) to the MyoChallenge Locomotion task. During Pre-training, a low-level policy comprised of 3 hidden layers with [1024, 1024, 512] units, which could

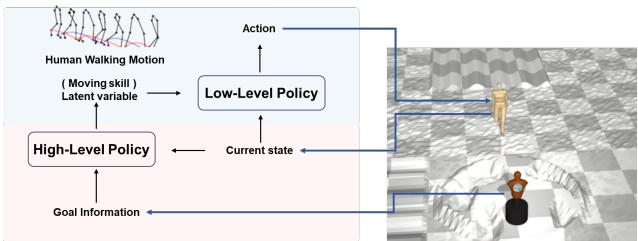

Figure 7: Two-stage framework (CarbonSiliconAI)

produce actions based on the current state and a latent variable representing a specific skill depicted

in human motion clips (including turn left, turn right, walk forward, walk backwards, slow down, speed up ...), and a discriminator that evaluated the realism of a motion were trained using PPO [19] and Adversarial Skill Embeddings (ASE) [30].

In consideration of the distinct skeleton differences between the MyoLeg and humans, the human walking clips provided by Adversarial Motion Priors ([26]) and Control-VAE ([31]), around 4 minutes, were retargeted to the MyoLeg's framework by straightforwardly mapping local joint rotations, root's scaled translation and orientation onto the MyoLeg's skeleton. After the low-level policy had the ability to perform life-like actions according to latent skills, a high-level policy was modeled using fully-connected network with 2 hidden layers of [1024, 512] units that took as input the current state and goal information, then specified latent to change the behaviors of the low-level policy to accomplish goals of chasing or evading. The high-level policy is trained using PPO [19] to satisfy a task reward while also trying to fool the discriminator by perform realistic behaviours that resemble motions shown in the human walking data. The state describes the configuration of MyoLeg, includes internal qpos, internal qvel, ground reaction force, torso angle, root position, root velocity, muscle length, muscle velocity, muscle force and action in the last time step. The goal information was comprised of task type (chase or evade), opponent linear speed, opponent rotation velocities, opponent position and opponent face direction in the MyoLeg's local frame.

# 4 Results

For phase 2, we computed standard deviations over 5000 episodes to differentiate potentially close scores.

## 4.1 Manipulation Track Results

During the first phase, Team Lattice obtained a score of $95.9\%$ with their methods in 3.1.1. During the second phase, Team lattice secure the winning place with a success rate of $33.5\% \pm 3\%$

The second place Team GaitNet achieves a perfect score ($100\%$) using the methods described in 3.1.2. In the second phase, GaitNet held the top spot on the leaderboard for a period, before achieving a final score of $32.3\% \pm 1\%$

In the first phase, Team CarbonSiliconAI obtained a score rate of $97\%$, with a final score of $21.5\% \pm 2\%$ in the second phase, with the methods described in 3.1.3

## 4.2 Locomotion Track Results

In the first phase, the winning team GaitNet obtained first place, with a success rate of $97\%$. They maintained their lead in the second phase, with a final score of $62.7\% \pm 4\%$, using the methods in 3.2.1

During the first phase, Team MSKBioDyn secured the second spot with a success rate of $61\%$ and $49\%$ in score. In the second phase, Team MSKBioDyn maintains its advantage by having a final score of $21.2\% \pm 3\%$, with the methods in 3.2.2

In the first phase, Team CarbonSiliconAI obtained a success rate of $36\%$ maintaining their position at 3rd place, with a final score of $13\% \pm 3\%$ in the second phase, using the methods described in 3.2.3

# 5 Discussions

## 5.1 Impact and Participation

This year's *MyoChallenge* had a total participation of 59 teams from over 15 countries. Across both phases, we had a total of 536 submissions. This widespread competition has also yielded remarkable results for MyoSuite, with over 6,000 total downloads during the competition phase, underlining its growing impact in the field. Additionally, 70% of the participants this year were newcomers, with

16.7% postgraduate researchers 50% graduate students, and one-third of master-level students. To promote diversity in science, we started a special DEI award for participants from an underrepresented population. We also have a Student award to promote participation among undergraduate students.

This competition was associated with a workshop at the NeurIPS 23 conference: MyoSymposium[5]. The MyoSymposium allowed us to bring together scholars and experts in the fields of biomechanics, ML, neuroscience, and health care.

## 5.2 Limitations and Lessons Learnt

**Lack of physiological accuracy**. Although we have seen great advancements in the use of machine learning to achieve both agility and dexterity in this edition of MyoChallenge, there is a lack of solutions arising from the biomechanical experts. All proposed solutions were based on reinforcement learning, which, while strong solutions, are limited due to their mismatch with human motor control mechanisms. Solutions inspired by fields other than machine learning could also help solve musculoskeletal control tasks. For example in lower-limb control, reflexes [32], sensorimotor connectivity priors [33] or central pattern generator [34] are simple yet extremely powerful solutions and they can create stable locomotion. Given the current reliance on imitating existing datasets of human movement, encouraging the creation of cross-disciplinary teams (biomechanics, neuroscience, and machine learning) that could facilitate the development of hybrid solutions is important for future challenges. One possible way to inspire such collaborations could be to provide biologically realistic sensory feedback, for example, with muscle spindles [35, 36], which might suffer from delays and incomplete information. This will bring us closer to the goal of understanding the human neurological control system.

**Underrepresented participation**. Another limitation was the small participation of an underrepresented population. For example, no participants came from South America or Africa in the past two challenges. Additionally, the involvement of women is low, with no winning teams containing women.

## 5.3 Future Challenges

**Promote participation in students**. Organizing such a large-scale event comes with numerous challenges requiring both technical e.g. setting up a website, helper code, infrastructure set-up and management, and logistical e.g. advertising and finding sponsors. Future challenges will promote the participation of students to help with different aspects of the technical and logistical planning and execution.

**Promote participation in underrepresented groups**. Additionally, we hope to lower the barriers to include researchers from underrepresented groups, underdeveloped countries, and students of all levels (e.g., high school, undergraduate, and master's). Efforts to achieve those goals include providing workshops and Q&A sessions throughout the challenge period and offering detailed tutorials and baseline code for newcomers of MyoSuite and MyoChallenge.

**Promote representation for people with limb loss**. Future editions of the *MyoChallenge* will be centered around the incorporation of bionic prosthetic limbs (both lower and upper) as part of a controller for dexterous motor tasks. Those topics would help explore how symbiotic human-robotic interaction needs to be coordinated to produce agile and dexterous behaviors. We hope to explore the opportunity to regain mobility and functionality for bionic limb human users and reclaim aspects of their former motor abilities.

---

[5]https://sites.google.com/view/myosuite/myochallenge/myochallenge-2023

## Acknowledgments and Disclosure of Funding

We would like to acknowledge support for this competition from the University of Twente Techmed and DSI, Northeastern University - The Institute for Experiential Robotics (IER), Google Cloud Computing and EU ERC StG Interact. Special thanks goes to Dhruv Batra, Ram Ramrakhya, Deshraj Yadav, and Rishabh Jain for help with the EvalAI platform. A.C., A.M.V. and A.M.: Swiss SNF grant (310030_212516)

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
