# OpenReview forum: "MyoChallenge 2023: Towards Human-Level Dexterity and Agility"
_NeurIPS.cc/2024/Datasets_and_Benchmarks_Track — Submitted to NeurIPS 2024 Track Datasets and Benchmarks_

### Official Review · Reviewer_tm9C · 2024-07-23
**More like a progress report rather than an innovative benchmark**

**Rating:** 5
**Confidence:** 4
**Correctness:** Correct
**Clarity:** Clear

**Review:**

The MyoChallenge 2023 stands out as a remarkable contribution to the field of neuromusculoskeletal research and artificial intelligence. The quality of the challenge is evidenced by its detailed and realistic simulation environment, which accurately mimics human arm and leg movements, providing a robust platform for testing and developing advanced control algorithms

**Strengths:**

One of the major strengths of the MyoChallenge 2023 is its highly realistic simulation environment. The use of detailed neuromusculoskeletal models for both arm and leg tasks provides a sophisticated platform that closely mimics real human movements.

By including a variety of tasks that increase in difficulty and require generalization under uncertain conditions, the challenge effectively assesses the robustness and versatility of participants' solutions.

**Additional Feedback:**

N/A

**Documentation:**

The challenges Starts on: Aug 9, 2022 5:00:00 PM PST (GMT - 7:00) and ends in 2099? Which is confusing. And it says: Myochallenge @ NeurIPS 2024 coming soon?

**Ethics:**

no ethics concerns.

**Limitations:**

no negative social impact

**Opportunities For Improvement:**

While the MyoChallenge 2023 builds upon the foundation laid by its predecessors in 2022 and earlier iterations, one might question the extent of its innovation given its continuation of established tracks.

The challenge's framework and objectives, while robust, largely follow the same lines as previous years. This repetition could be seen as a lack of groundbreaking novelty, as the core principles and methodologies remain consistent.

Additionally, while the tasks have become incrementally more complex, they still revolve around similar themes of manipulation and locomotion, potentially limiting the scope for new and diverse applications.

To truly push the boundaries of innovation, future challenges might need to introduce entirely new paradigms or significantly expand the variety of tasks to explore uncharted territories in AI and robotics.

**Relation To Prior Work:**

Addressed

**Summary And Contributions:**

The MyoChallenge 2023 aimed to push the boundaries of human-level dexterity and agility by featuring two main tracks: Manipulation and Locomotion. Both tracks involved using neuromusculoskeletal models to mimic human movements in complex tasks.

The Manipulation Track requires participants to control a 63-musculoskeletal arm model to maneuver objects of varying shapes and sizes.
The Locomotion Track involves an 80-musculoskeletal leg model where agents had to chase or evade a moving object.

Both tracks highlighted the overarching theme of dexterity and agility, and the Myosuite framework facilitated these challenges by providing a realistic, contact-rich, and computation-efficient virtual model.

---

> ### Author Rebuttal · Authors · 2024-08-16
>
> We thank the reviewer for their feedback. We are happy to read that the MyoChallenge is seen as a remarkable contribution to the field and that the variety of tasks and difficulty of the benchmark is noted.
>
> >The challenge's framework and objectives, while robust, largely follow the same lines as previous years. This repetition could be seen as a lack of groundbreaking novelty, as the core principles and methodologies remain consistent.
>
> We agree that the core principles and methodologies remain the same as before. A familiar setup and scoring system was chosen to build a sense of familiarity with previous participants. Many people from previous years have reached out to us about new iterations of the challenge and our surveys have also shown that, while we had many newcomers, there is also a growing number of repeat participants.
>
> Nevertheless, the 2023 tracks feature new simulation models which were not present in the first iteration of the challenge. For 2022, only a musculoskeletal hand model (39 muscles) and 2 different objects were used, while 2023 presents a musculoskeletal arm with shoulder and fully featured hand (63 muscles), as well as a completely new bipedal human model.
> While some iterations of the previous OpenSim-based NeurIPS challenges already featured locomotion models, none of them had a humanoid of this complexity (80 muscles). Hence, the complexity of the control required for this challenge was significantly greater than any of the challenges before.
>
> Additionally, while the tasks have become incrementally more complex, they still revolve around similar themes of manipulation and locomotion, potentially limiting the scope for new and diverse applications.
> In contrast to the previous year, both tracks required a very strong generalization component. The manipulation track presented the agents with unknown objects during evaluation, which participants could not prepare for. The locomotion track featured not only a randomly changing terrain map, unseen to this extent in the previous OpenSim NeurIPS challenges, but also requires high-level reasoning in addition to low-level control. This level of reasoning was unseen in other musculoskeletal challenges.
>
> > To truly push the boundaries of innovation, future challenges might need to introduce entirely new paradigms or significantly expand the variety of tasks to explore uncharted territories in AI and robotics.
>
> We strongly agree with the reviewer on this aspect. The MyoChallenge is about pushing the field towards uncharted territories.
> We want to highlight that the winning solutions featured approaches combining imitation learning with pure RL in order to manage the generalization aspect and the complexity of the tasks. This goes vastly beyond the mere application of existing RL algorithms to the tasks.
>
> To push the field even further, the 2024 edition of the challenge contains active bionic limbs that need to be co-controlled with the musculoskeletal system.
>
>
> > The challenges Starts on: Aug 9, 2022 5:00:00 PM PST (GMT - 7:00) and ends in 2099? Which is confusing. And it says: Myochallenge @ NeurIPS 2024 coming soon?
>
> We want to point to our extensive tutorials and online documentation for the challenge:
> https://github.com/MyoHub/myochallenge_2023eval
>
>
> The ending date of 2099 was chosen such that the web backend for the challenge accepts further submissions and can act as an indefinite benchmark. The real challenge ending date was communicated through other channels.
>
> >  And it says: Myochallenge @ NeurIPS 2024 coming soon?
>
> We are unsure what exactly the reviewer is refering to. However, this might have been in reference to the next challenge, which will be featured as a competition for NeurIPS 2024, and not to the current submission, which is detailing the 2023 competition.

---

### Official Review · Reviewer_MbhK · 2024-07-24
**interesting control benchmarks but paper itself could be improved**

**Rating:** 5
**Confidence:** 3

**Review:**

The models are novel and provide challenging control problems. The suite is well-documented.

I was in the borderline area for this paper because although the benchmark looks good and useful, the paper itself appears a bit scrappy, with relatively little technical detail on the challenges themselves and the evaluation mechanisms, and a lot of discursive content about the solution strategies. For a Datasets and Benchmarks paper it would seem more important to make sure the benchmark itself was well documented.

**Strengths:**

The models are novel and provide challenging, high-dimensional control problems which are relevant for control, robotics, rehabilitation engineering. The suite is well-documented.

**Additional Feedback:**

n/a

**Clarity:**

The paper itself appears a bit scrappy, with relatively little technical detail on the challenges themselves and the evaluation mechanisms, and a lot of discursive content about the solution strategies. For a Datasets and Benchmarks paper it would seem more important to make sure the benchmark itself was well documented.

**Correctness:**

it appears to be documented well enough to have supported multiple teams at a workshop, so I assume that the major issues have been ironed out.

**Documentation:**

it appears to be documented well enough to have supported multiple teams at a workshop, so I assume that the major issues have been ironed out.

**Ethics:**

nothing specific

**Limitations:**

The limitations section as it currently is seems to focus on the limitations of the solutions attempted, rather than the benchmark itself. There is a suggestion of how to improve the sensory feedback, which is relevant, but I think more detailed discussion of the limitations of the simulation and evaluation metrics would be appropriate.

**Opportunities For Improvement:**

I was in the borderline area for this paper because although the benchmark looks good and useful, the paper itself appears a bit scrappy, with relatively little technical detail on the challenges themselves and the evaluation mechanisms, and a lot of discursive content about the solution strategies. For a Datasets and Benchmarks paper it would seem more important to make sure the benchmark itself was well documented.

Adding some human-generated data for some of these tasks would seem like a valuable addition to the current benchmarks (would need some intelligent ways to map different measured states to those of the simulation, but would add an interesting benchmark comparison and let people check their solutions for humanlike characteristics). This would help address the point in Section 5.2 about the lack of physiological accuracy.

**Relation To Prior Work:**

a fairly good and uptodate literature review is included.

**Summary And Contributions:**

the paper presents the MyoChallenge2023 tasks, and evaluation metrics, and gives a summary of the leading solutions from the 2023 workshop at Neurips. The challenge represents a novel executable simulation of a high-dimensional, dynamic system with simulated skin sensing, which can be used to evaluate different control strategies.

---

> ### Author Rebuttal · Authors · 2024-08-17
>
> > ... with relatively little technical detail on the challenges themselves and the evaluation mechanisms, and a lot of discursive content about the solution strategies. For a Datasets and Benchmarks paper it would seem more important to make sure the benchmark itself was well documented.
>
> Thank you for your feedback. Our goal with this report was to present both the challenge benchmark and the scientific achievements obtained by the participants in solving these complex tasks. We understand that, given the topic of a data and benchmark track, greater emphasis on the benchmark would be preferred. We will revise the description to include more discussions on the limitations of both the simulation and the evaluation metrics. Below is a more detailed description of the challenge.
>
> For both challenges, the main objective is to encourage participants to develop policies that can control complex models and generalize well in their respective tasks.
>
> In the Manipulation track, the challenge features a biomechanically accurate human musculoskeletal full arm (63 muscles, 27 DOF), compared to the musculoskeletal hand model (39 muscles, 23 DOF) from the previous edition in 2022. The new musculoskeletal model allows for greater task complexity by introducing shoulder joints that rotate and translate simultaneously during movement.
>
> Observations provided for this challenge were:
> 1. Current hand position
> 2. Current hand velocity
> 3. Object position
> 4. Object orientation
> 5. Goal position
> 6. Goal orientation
> 7. Object position error (in relation to goal position)
> 8. Object orientation error (in relation to goal orientation)
>
> Participants were free to use all or a subset of the observation when developing their policies.
>
> During evaluation, the object position and rotation error (relative to the goal) is continuously checked until the task is deemed successful or the task horizon is reached. Task success is defined as having the object position and orientation error below a threshold. Participants receive a positive score of 1 when the task is successful; otherwise, it is 0. The effort metric, represented as the squared sum of muscle activations, encourages participants to develop energy-efficient policies.
>
> In the Locomotion track, a new musculoskeletal leg (MyoLeg) model with 80 muscles and 16 internal DOF was introduced. In addition to locomotion, participants must develop policies to chase and evade a moving target over different types of terrain. This task encompasses multiple locomotion tasks, requiring policies to efficiently combine running, chasing, evading, and handling various types of rough terrain
>
> Observations provided for this challenge were:
> 1. Internal joint positions
> 2. Internal joint velocities
> 3. Ground reaction forces
> 4. Torso angles
> 5. Muscle length
> 6. Muscle velocities
> 7. Muscle forces
> 8. Muscle activations
> 9. Self world position
> 10. Self world velocity
> 11. Opponent world position
> 12. Opponent world velocity
> 13. 10x10 heightfield grid
>
> As with the Manipulation track, participants were free to use all or part of the observations provided. Most of the observations are self-explanatory, except for the 10x10 heightfield grid, which provides information about the terrain heightfield around the musculoskeletal model, akin to a visual representation of the terrain.
>
> During evaluation, policies were assessed based on the combined success rates, scores, and effort of two tasks: Chase and Evade. For Chasing, policies must move the MyoLeg model within a set radius of the target within 20 seconds. For Evading, policies must keep the MyoLeg outside the same radius around the target. Successful episodes earn 1 point; otherwise, they earn 0. The score metric (Section 2.4) measures the efficiency of a policy in both tasks. A higher score indicates that the policy takes less time to chase and survives longer in evasion. Similar to the Manipulation track, the effort metric evaluates how energy efficient the policy is.
>
> > I think more detailed discussion of the limitations of the simulation and evaluation metrics would be appropriate.
>
> One limitation of the locomotion simulation is the constant terrain friction, which may not be realistic. However, since the task is modeled after the real-world competition, ChaseTag, we assume the friction coefficients of the competition arena are relatively constant, corresponding to an even, stationary ground. Future improvements could include more natural ground for better generalization. In the manipulation simulation, the lack of vision and proprioception makes the task challenging for generalization, as the hand can only adjust the object’s shape during manipulation while it slides away, resulting in unnatural grasping. Incorporating sensory information, such as touch, would be needed for more naturalistic generalization to object shapes. Additionally, position error metrics do not guarantee that the controlling policy will not “throw” the object towards the goal, limiting success chances, and this approach was not adopted by competitors. For locomotion, combining the score metric for both chase and evade might not properly evaluate the trained policy’s capabilities. Different policies could achieve similar scores with varying strengths in chase and evade tasks. For example, Policy A might excel in chase but perform poorly in evade, while Policy B performs reasonably well in both tasks, yet both could end up with similar scores.
>
> > Adding some human-generated data for some of these tasks would seem like a valuable addition to the current benchmarks
>
> We agree that human data would be valuable, however, given that such data tend to be idiosyncratic, it is difficult to design proper metrics to accurately capture the variability of the tasks. Nevertheless, we have considered this point and have already reached out to several groups to solicitate data and metrics of humans using prosthetics for our upcoming MyoChallenge 2024.

---

> > ### Comment · Reviewer_MbhK · 2024-08-26
> > **thank you for the additional background information**
> >
> > thank you for the additional background information about the benchmark.

---

### Official Review · Reviewer_t2QP · 2024-07-25
**Review of the paper**

**Rating:** 8
**Confidence:** 3
**Correctness:** The claims made in the submission is …
**Clarity:** Yes, the paper is well written.

**Review:**

The submission demonstrates strong methodological rigor in the development and execution of the MyoChallenge. It utilizes MyoSuite 2, a computationally efficient framework, to simulate complex musculoskeletal models accurately. Tasks in the Manipulation and Locomotion tracks are well-defined and progressively challenging across phases, ensuring a thorough evaluation of participant performance.

While the Manipulation track showed high performance in phase 1, the Locomotion track revealed variability, especially in phase 2, indicating potential limitations in generalization across diverse conditions.

**Strengths:**

1, The submission addresses fundamental questions about human motor control by leveraging state-of-the-art simulations and machine learning techniques. It aims to bridge gaps in current models by striving to match human-level dexterity and agility.

2, By developing models that simulate the complexities of human musculoskeletal systems, the MyoChallenge contributes to fields such as neuroscience, biomechanics, robotics, and rehabilitation. This has direct implications for improving assistive technologies and rehabilitation strategies.

3, By bringing together experts from biomechanics, machine learning, neuroscience, and physics simulators, the MyoChallenge promotes interdisciplinary collaboration. This facilitates the integration of diverse methodologies and expertise to tackle complex problems in motor control.

4, The challenge provides a benchmarking platform (MyoSuite 2) that supports efficient and accurate simulations of musculoskeletal models. This not only fosters innovation within the community but also sets standards for future research and development in motor control.

**Additional Feedback:**

None

**Documentation:**

Yes, it is sufficient detail on Documentation.

**Limitations:**

The same as the Opportunities For Improvement.

**Opportunities For Improvement:**

1, There are no real-world experiments, with limit the contribution of this work.
2, The manipulation task is relatively simple.

**Relation To Prior Work:**

The paper clearly discussed the related work.

**Summary And Contributions:**

The paper aimed to explore human-like dexterity and agility through two tracks: Manipulation and Locomotion, using a realistic virtual neuromusculoskeletal model. Participants worked with a 63-musculoskeletal arm model for Manipulation, focusing on stable grasps of various-shaped objects, and an 80-musculoskeletal model for Locomotion, requiring agents to navigate by chasing or evading a moving object.

---

> ### Author Rebuttal · Authors · 2024-08-16
>
> We thank the reviewers for their insightful suggestions.
>
> >1. There are no real-world experiments, with limit the contribution of this work.
>
> We would like to first point out that the musculoskeletal models used in MyoSuite, including their muscle, tendon, and bone geometries, are grounded in experimental data and studies (Xu et al. (2012) for Finger, Rajagopal et al. (2016), Holzbaur et al (2005) for the arm).
> Regarding the comparison of our models with physiological experimental data in this competition, there are several challenges.  First, capturing data simultaneously from all 63 muscles involved in manipulation and 80 muscles in locomotion, together with contact forces on objects as simulated in our model, is currently not feasible due to extensive human and resource requirements. Second, data from individual human subjects tend to be idiosyncratic and do not reliably represent the variability required for the broad range of tasks we are investigating.
> Nevertheless, we agree that including some experimental data could improve the benchmarking and evaluation of data-driven methods, potentially through the use of advanced imitation learning or inverse RL. Hence, we have incorporated a metric for physiological accuracy in the locomotion tasks for the forthcoming MyoChallenge '24.
>
> >2. The manipulation task is relatively simple.
>
> We thank the reviewer for feedback on the manipulation task.
> Although the manipulation track might seem relatively simple at first glance, it's important to consider the complexity introduced by the musculoskeletal arm, with a shoulder and fully functional hand, comprising 63 muscles. The dexterity required to manipulate an object involves a control sophistication never before achieved. Additionally, during the second phase of the competition, the trained policy demanded a significant generalization component. Specifically, the manipulation track challenged participants by presenting unknown objects during evaluation, which could not be anticipated or prepared for in advance. This added a substantial layer of complexity to the task. In addition, the winning solution of the manipulation track only reaches ~35%, suggesting room for improvement and highlighting the challenge in generalization in the face of the placement task.
> To further push the boundaries of the field, in this year's MyoChallenge, we increased the complexity of the task by introducing a bimanual control task with a prosthesis. We appreciate your insights and hope this clarifies the nature of the challenges faced in the competition.

---

### Decision · Program_Chairs · 2024-09-26

**Decision:**

Reject

**Comment:**

This paper is a competition analysis paper submitted to the D&B track. The paper presented the continuation of the MyoChallenge challenge, including two tracks of Manipulation and Locomotion, presented competition results and solution strategies. The paper got mixed reviews with one reviewer strongly advocate the importance of the work while two other reviewers have clear conservations regarding the presentation of the paper, and the significance of improvement over the same challenge hosted in prior year. The AC has carefully read the paper and all messages. While the authors' rebuttal has addressed both concerns to some extent, it appears to the AC that a serious round of paper re-writing is necessary to make the paper pass the acceptance bar of the track. It is a hard decision, but the AC has to decide not recommend an acceptance for the work in its current form.